# Adversarial Examples Make Strong Poisons

**Liam Fowl**[*]
Department of Mathematics
University of Maryland
lfowl@umd.edu

**Micah Goldblum**[*]
Department of Computer Science
University of Maryland

**Ping-yeh Chiang**[*]
Department of Computer Science
University of Maryland

**Jonas Geiping**
Department of Electrical Engineering
University of Siegen

**Wojtek Czaja**
Department of Mathematics
University of Maryland

**Tom Goldstein**
Department of Computer Science
University of Maryland

## Abstract

The adversarial machine learning literature is largely partitioned into evasion attacks on testing data and poisoning attacks on training data. In this work, we show that adversarial examples, originally intended for attacking pre-trained models, are even more effective for data poisoning than recent methods designed specifically for poisoning. Our findings indicate that adversarial examples, when assigned the original label of their natural base image, cannot be used to train a classifier for natural images. Furthermore, when adversarial examples are assigned their adversarial class label, they are useful for training. This suggests that adversarial examples contain useful semantic content, just with the "wrong" labels (according to a network, but not a human). Our method, *adversarial poisoning*, is substantially more effective than existing poisoning methods for secure dataset release, and we release a poisoned version of ImageNet, ImageNet-P, to encourage research into the strength of this form of data obfuscation.

## 1 Introduction

Automated dataset scraping has become necessary to satisfy the exploding demands of cutting-edge deep models [Bonawitz et al., 2019, Brown et al., 2020], but the same automation that enables massive performance boosts exposes these models to security vulnerabilities [Bagdasaryan et al., 2020, Chen et al., 2020]. *Data poisoning* attacks manipulate training data in order to cause the resulting models to misclassify samples during inference [Koh and Liang, 2017], while *backdoor attacks* embed exploits which can be triggered by pre-specified input features [Chen et al., 2017]. In this work, we focus on a flavor of data poisoning known as *availability attacks*, which aim to degrade overall testing performance [Barreno et al., 2010, Biggio et al., 2012].

Adversarial attacks, on the other hand, focus on manipulating samples at test-time, rather than during training [Szegedy et al., 2013]. In this work, we connect adversarial and poisoning attacks by showing that adversarial examples form stronger availability attacks than any existing poisoning method, even though the latter were designed specifically for manipulating training data while adversarial examples were not. We compare our method, *adversarial poisoning*, to existing availability attacks for neural

---

[*] Authors contributed equally.

networks, and we exhibit consistent performance boosts (i.e. lower test accuracy). In fact, models trained on adversarial examples may exhibit test-time performance below that of random guessing.

Intuitively, adversarial examples look dramatically different from their natural base images in the eye of neural networks, despite the two looking similar to humans. Models trained only on such perturbed examples are completely unprepared for inference on the clean data. In support of this intuition, we observe that models trained on adversarially perturbed training data often fail to correctly classify the original clean training samples.

But does this phenomenon occur simply because adversarial examples are off the "natural image manifold" or because they actually contain informative features from other classes? Popular belief assumes that adversarial examples live off the natural image manifold, causing a catastrophic mismatch when digested by models trained on only clean data [Khoury and Hadfield-Menell, 2018, Stutz et al., 2019, Zhao et al., 2017]. However, models trained on data with random additive noise (rather than adversarial noise) perform well on noiseless data, suggesting that there may be more to the effects of adversarial examples than simply moving off the manifold (see Table 2). We instead find that since adversarial attacks inject features that a model associates with incorrect labels, training on these examples is similar to training on *mislabeled* training data. After re-labeling adversarial examples with the "wrong" prediction of the network on which they were crafted, models trained on such label-flipped data perform substantially better than models trained on uncorrected adversarial examples and almost as well as models trained on clean images. While this label-correction is infeasible for a practitioner defending against adversarial poisoning, since it assumes possession of the crafting network which requires access to the clean dataset, this experiment strengthens the intuition that adversarial examples contain a strong training signal just from the "wrong" class.

## 2   Related Work

Data poisoning can generally be phrased as a bilevel optimization problem which minimizes loss with respect to parameters in the inner problem while maximizing some attack loss with respect to inputs in the outer problem [Biggio et al., 2012, Huang et al., 2020a]. Poisoning comes in several flavors, including *integrity attacks* and *availability attacks*. The former aims to cause targeted misclassification on a small number of pre-selected datapoints, while the latter aims to degrade the overall performance (generalization ability) of a victim network Barreno et al. [2010]. Classical approaches to poisoining attacks often focused on simple models, where the inner problem can sometimes be solved exactly [Biggio et al., 2012, Mei and Zhu, 2015, Xiao et al., 2015, Chan-Hon-Tong, 2019]. However, on neural networks, obtaining exact solutions is intractable. In this setting, Muñoz-González et al. [2017] approximates a solution to the inner problem using a small number of descent steps, but the authors note that this method is ineffective against deep neural networks.

Modern poisoning attacks adopt new methods and approximations, like gradient alignment [Geiping et al., 2020, Souri et al., 2021], computation graph unrolling [Huang et al., 2020b], etc., but for integrity attacks on deep networks. On the availability attack side, still other heuristics have been adopted. For example, gradient alignment [Geiping et al., 2020] with a modified indiscriminate objective was used in Fowl et al. [2021], while gradient explosion was suggested in Shen et al. [2019]. Other availability attacks have used auto-encoder generated perturbations [Feng et al., 2019a], as well as loss minimization objectives [Huang et al., 2021]. Notably, it was previously believed that adversarial (loss maximization) objectives were *not* suitable to availability attacks [Huang et al., 2021].

Still other related poisoning works harness *influence functions* which estimate the impact of each training sample on a resulting model [Fang et al., 2020, Koh and Liang, 2017, Koh et al., 2018]. However, influence functions are brittle on deep networks whose loss surfaces are highly irregular [Basu et al., 2020]. A general overview of data poisoning methods can be found in Goldblum et al. [2020].

**Adversarial examples.** Adversarial attacks probe the blindspots of trained models where they catastrophically misclassify inputs that have undergone small perturbations [Szegedy et al., 2013]. Prototypical algorithms for adversarial attacks simply maximize loss with respect to the input while constraining perturbations. The resulting adversarial examples exploit the fact that as inputs are even slightly perturbed in just the right direction, their corresponding deep features and logits change dramatically, and gradient-based optimizers can efficiently find these directions. The literature

contains a wide array of proposed loss functions and optimizers for improving the effectiveness of attacks [Carlini and Wagner, 2017, Gowal et al., 2019]. A number of works suggest that adversarial examples are off the image manifold, and others propose methods for producing on-manifold attacks [Khoury and Hadfield-Menell, 2018, Stutz et al., 2019, Zhao et al., 2017].

**Adversarial training.** The most popular method for producing neural networks that are robust to attacks involves crafting adversarial versions of each mini-batch and training on these versions [Madry et al., 2017]. On the surface, it might sound as if adversarial training is very similar to training on poisons crafted via adversarial attacks. After all, they both involve training on adversarial examples. However, adversarial training ensures that the robust model classifies inputs correctly within a ball surrounding each training sample. This is accomplished by updating perturbations to inputs *throughout* training. This process desensitizes the adversarially trained model to small perturbations to its inputs. In contrast, a model trained on adversarially poisoned data is only encouraged to fit the exact, fixed perturbed data.

# 3 Adversarial Examples as Poisons

## 3.1 Threat Model and Motivation

We introduce two parties: the *poisoner* (sometimes called the attacker), and the *victim*. The poisoner has the ability to perturb the victim's training data but does not know the victim's model initialization, training routine, or architecture. The victim then trains a new model from scratch on the poisoned data. The poisoner's success is determined by the accuracy of the victim model on clean data.

Early availability attacks that worked best in simple settings, like SVMs, often modified only a small portion of the training data. However, recent availability attacks that work in more complex settings have instead focused on applications such as secure data release where the poisoner has access to, and modifies, *all* data used by the victim [Shen et al., 2019, Feng et al., 2019a, Huang et al., 2021, Fowl et al., 2021].

These methods manipulate the entire training set to cause poor generalization in deep learning models trained on the poisoned data. This setting is relevant to practitioners such as social media companies who wish to maintain the competitive advantage afforded to them by access to large amounts of user data, while also protecting user privacy by making scraped data useless for training models. Practically speaking, companies could employ methods in this domain to imperceptibly modify user data *before* dissemination through social media sites in order to degrade performance of any model which is trained on this disseminated data.

To compare our method to recent works, we focus our experiments in this setting where the poisoner can perturb the entire training set. However, we also poison lower proportions of the data in Tables 5, 4. We find that on both simple and complex datasets, our method produces poisons which are useless for training, and models trained on data including poisons would have performed just as well had they identified and thrown out the poisoned data altogether.

## 3.2 Problem Setup

Formally stated, availability poisoning attacks aim to solve the following bi-level objective in terms of perturbations $\delta = \{\delta_i\}$ to elements $x_i$ of a dataset $\mathcal{T}$:

$$\max_{\delta \in \mathcal{S}} \ \mathbb{E}_{(x,y) \sim \mathcal{D}} \left[ \mathcal{L}\left(F(x; \theta(\delta)), y\right) \right] \tag{1}$$

$$\text{s.t. } \theta(\delta) \in \arg\min_{\theta} \sum_{(x_i, y_i) \in \mathcal{T}} \mathcal{L}(F(x_i + \delta_i; \theta), y_i), \tag{2}$$

where $\mathcal{S}$ denotes the constraint set of the perturbations. As is common in both the adversarial attack and poisoning literature, we employ an $\ell_\infty$ bound on each $\delta_i$. Unless otherwise stated, our attacks are bounded by $\ell_\infty$-norm $\epsilon = 8/255$ as is standard practice on CIFAR-10, and ImageNet data in both adversarial and poisoning literature [Madry et al., 2017, Geiping et al., 2020]. Simply put, the attacker wishes to cause a network, $F$, trained on the poisons to generalize poorly to distribution $\mathcal{D}$ from which $\mathcal{T}$ was sampled.

Directly solving this optimization problem is intractible for neural networks as it requires unrolling the entire training procedure found in the inner objective (Equation (2)) and backpropagating through it to perform a single step of gradient descent on the outer objective. Thus, the attacker must approximate the bilevel objective. Approximations to this objective often involve heuristics, as previously described. For example, TensorClog [Shen et al., 2019] aims to cause gradient vanishing in order to disrupt training, while more recent work aims to align poison gradients with an adversarial objective [Geiping et al., 2020].

We opt for an entirely different strategy and instead replace the bi-level problem with *two* empirical *loss maximization* problems - an approach that was believed to be suboptimal for availability poisoning [Huang et al., 2021]. This turns the poison generation problem into an adversarial example problem. Specifically, we optimize the following untargeted (UT) objective:

$$\max_{\delta \in \mathcal{S}} \left[ \sum_{(x_i, y_i) \in \mathcal{T}} \mathcal{L}\left(F(x_i + \delta_i; \theta^*), y_i\right) \right], \tag{3}$$

where $\theta^*$ denotes the parameters of a model trained on *clean* data, which is fixed during poison generation. We call this model the *crafting* model.

We also optimize an objective which defines a *class targeted* (CT) adversarial attack. This modified objective is defined by:

$$\min_{\delta \in \mathcal{S}} \left[ \sum_{(x_i, y_i) \in \mathcal{T}} \mathcal{L}\left(F(x_i + \delta_i; \theta^*), g(y_i)\right) \right], \tag{4}$$

where $g$ is a permutation (with no fixed points) on the label space of $\mathcal{S}$. Fittingly, we call our methods *adversarial poisoning*. Note that the class targeted objective was previously (independently) hypothesized to produce potent poisons in Nakkiran [2019], and also tested in a work concurrent to ours [Tao et al., 2021].

Projected Gradient Descent (PGD) has become the standard method for generating adversarial examples for deep networks [Madry et al., 2017]. Accordingly, we craft our poisons with 250 steps of PGD on this loss-maximization objective. In addition to the adversarial attack introduced in Madry et al. [2017], we also experiment with other attacks such as FGSM [Goodfellow et al., 2014] and Carlini-Wagner [Carlini and Wagner, 2017] in Appendix Table 2. We find that while other adversaries do produce effective poisons, a PGD based attack is the most effective in generating poisons. Finally, borrowing from recent targeted data poisoning works, we also employ differentiable data augmentation in the crafting stage [Geiping et al., 2020] (see section 3.3).

An aspect of note for our method is the ease of crafting perturbations - we use a straightforward adversarial attack on a fixed pretrained network to generate the poisons. This is in contrast to previous works which require pretraining an adversarial auto-encoder [Feng et al., 2019b] (5 - 7 GPU days for simple datasets), or require iteratively updating the model and perturbations [Huang et al., 2021], which requires access to the entire training set all at once - an assumption that does not hold for practitioners like social media companies who acquire data sequentially. In addition to the performance boosts our method offers, it is also the most flexible compared to each of the availability attacks with which we compare.

### 3.3 Technical Details for Successful Attacks

As we will see in the following sections, adversarial objectives can indeed produce powerful poisons. As previously stated, such loss maximization approaches to availability attacks were thought to be suboptimal [Huang et al., 2021]. However, we find that differentiable data augmentation during crafting, along with more PGD steps, greatly improves the potency of the generated poisons. In Appendix Fig. 6, we see that augmentation is very helpful for the untargeted objective at higher number of PGD steps, but is less important for the class-targeted objective.

Furthermore, we find that the variability of the untargeted attack is generally higher than the class-targeted attack. That is, the untargeted attack is more sensitive to random initialization of the poisons. For purposes of comparison, we do modest hyperparameter searching to find a reasonable poisoned dataset on which to evaluate a victim model. We discuss this further in Appendix Sec. A.9.

## 3.4 Baseline CIFAR-10 Results

We first experiment in a relatively simple setting, CIFAR-10 [Krizhevsky et al., 2009], consisting of $50,000$ low-resolution images from 10 classes. All poisons in this setting are tested on a variety of architectures in a setting adapted from a commonly used repository [2]. We find that our poisoning method reliably degrades the test accuracy of a wide variety of popular models including VGG19 [Simonyan and Zisserman, 2014], ResNet-18 [He et al., 2015], GoogLeNet [Szegedy et al., 2015], DenseNet-121 [Huang et al., 2016], and MobileNetV2 [Sandler et al., 2018]. Even under very tight $\ell_\infty$ constraints, our poisons more than halve the test accuracy of these models. These results are presented in Table 1.

Table 1: **Comparison of different $\varepsilon$-bounds for our adversarial poisoning method.** All poisons generated by ResNet-18 crafted with 250 steps of PGD, with differentiable data augmentation. CT denotes poisons crafted with the class targeted adversarial attack. Note that we would expect random guessing to achieve $10\%$ validation accuracy.

| BOUND \ VICTIM | VGG19 | RESNET-18 | GOOGLENET | DENSENET-121 | MOBILENETV2 |
|---|---|---|---|---|---|
| CLEAN | $93.9 \pm 0.16$ | $95.53 \pm 0.03$ | $95.38 \pm 0.11$ | $95.51 \pm 0.07$ | $92.42 \pm 0.06$ |
| $\varepsilon = 4/255$ (UT) | $64.71 \pm 0.76$ | $56.79 \pm 0.75$ | $61.9 \pm 0.42$ | $59.10 \pm 0.34$ | $47.72 \pm 0.23$ |
| $\varepsilon = 8/255$ (UT) | $10.98 \pm 0.27$ | $6.25 \pm 0.17$ | $7.03 \pm 0.12$ | $7.16 \pm 0.16$ | $6.11 \pm 0.17$ |
| $\varepsilon = 4/255$ (CT) | $30.26 \pm 0.55$ | $26.49 \pm 0.15$ | $29.53 \pm 0.51$ | $26.63 \pm 0.59$ | $21.64 \pm 0.54$ |
| $\varepsilon = 8/255$ (CT) | $10.32 \pm 0.35$ | $8.69 \pm 0.44$ | $9.36 \pm 0.26$ | $7.85 \pm 0.40$ | $8.36 \pm 0.31$ |

Additionally, we study the effect of the optimizer, data augmentation, number of steps, and crafting network on poison generation in Appendix Tables 10, 11, and 12. These tables demonstrate that a wide variety of adversarial attacks with various crafting networks and hyperparameters yield effective poisons. Also note that while the results we present here are for poisons generated with a ResNet18, we find that the poisons remain effective when generated from other network architectures (see Appendix Table 12).

In this popular setting, we can compare our method to existing availability attacks including Tensor-Clog [Shen et al., 2019], Loss Minimization [Huang et al., 2021], and a gradient alignment based method [Fowl et al., 2021]. Our method widely outperforms these previous methods. Compared with the previous best method (loss minimization), we degrade the validation accuracy of a victim network by a factor of more than three.

Table 2: **Validation accuracies of models trained on data from different availability attacks.** Tested on randomly initialized ResNet-18 models on CIFAR-10. All crafted with $\varepsilon = 8/255$.

| METHOD | VALIDATION ACCURACY ($\%, \downarrow$) |
|---|---|
| NONE (CLEAN) | 94.56 |
| RANDOM NOISE | 90.52 |
| TENSORCLOG [SHEN ET AL., 2019] | 84.24 |
| ALIGNMENT [FOWL ET AL., 2021] | 53.67 |
| UNLEARNABLE EXAMPLES [HUANG ET AL., 2021] | 19.85 |
| DEEPCONFUSE [FENG ET AL., 2019B] | 31.10 |
| ADVERSARIAL POISONING UNTARGETED (OURS) | **11.94** |
| ADVERSARIAL POISONING CLASS-TARGETED (OURS) | **8.69** |

Note that we test our method in a completely black-box setting wherein the poisoner has no knowledge of the victim network's initialization, architecture, learning rate scheduler, optimizer, etc. We find our adversarial poisons transfer across these settings and reliably degrade the validation accuracy of all the models tested. See Appendix A.1 for more details about training procedures.

## 3.5 Large Scale Poisoning

In addition to validating our method on CIFAR-10, we also conduct experiments on ImageNet (ILSVRC2012), consisting of over 1 million images coming from 1000 different classes [Russakovsky

---

[2]`https://github.com/kuangliu/pytorch-cifar`.

et al., 2015]. This setting tests whether adversarial poisons can degrade accuracy on industrial-scale, high resolution datasets. We discover that while untargeted attacks successfully reduce the generalization, our class-targeted attack (CT) degrades clean validation accuracy significantly further (see Table 3). Furthermore, even at an almost imperceptible perturbation level of $4/255$, the class targeted poisons cripple the validation accuracy of the victim model to $3.57\%$. Visualizations of the poisons can be found in Figure 1.

Table 3: **Validation accuracies of models trained on poisoned ImageNet data**. Tested on randomly initialized ResNet-18 models.

| METHOD | VALIDATION ACCURACY ($\%, \downarrow$) |
|---|---|
| NONE (CLEAN) | 66.56 |
| $\varepsilon = 4/255$ (UT) | 45.07 |
| $\varepsilon = 8/255$ (UT) | 36.63 |
| $\varepsilon = 4/255$ (CT) | 3.57 |
| $\varepsilon = 8/255$ (CT) | **1.45** |

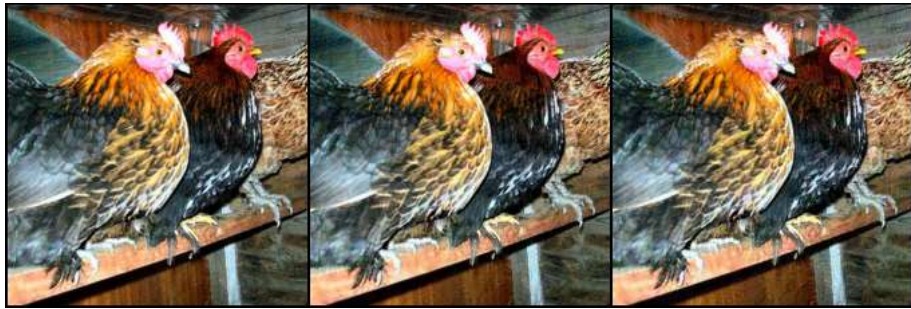

Figure 1: Randomly selected clean image (left), with perturbed counterparts at level $\varepsilon = 4/255$ (center), and $\varepsilon = 8/255$ (right). The clean image is taken from class "hen" and poisons are generated via perturbations into class "ostrich".

We hypothesize that the noteable superiority of the class targeted attack in this setting arises from the larger label-space of ImageNet. Specifically, we find that untargeted adversarial attacks can result in a concentrated attack matrix - i.e. attacks perturb data into a relatively small number of classes. This could lead to the perturbations being less effective at degrading generalization because they are not discriminatively useful for a network and are thus ignored. In contrast, a class-targeted attack ensures that the adversarially perturbed features can be associated with an incorrect class label when learned by a victim network.

## 3.6 Facial Recognition

In our third experimental setting, we test whether our poisons are effective against facial recognition models, along the lines of the motivations introduced in Section 3.1, where companies like social media sites could employ our method to protect user data from being scraped and used by facial recognition networks as demonstrated in [Cherepanova et al., 2021]. For the purposes of comparison, we constrain our attack to the same setting as Huang et al. [2021]. In this setting, we only assume the ability to modify a subset of the face images, specifically, the face images of the users that we want to protect. We test on the Webface dataset, and use pretrained Webface model from Yi et al. [2014] to generate the class targeted adversarial examples with $\varepsilon = 8/255$. Our class targeted adversarial examples are exceedingly effective in this domain. Our method reduces the protected classes' accuracy down to $8\%$, half the classification success of the unlearnable examples method of Huang et al. [2021] (see Table 4). We visualize perturbations in Appendix Figure 5. Note that these images are produced using the same $\varepsilon$ bound as unlearnable examples for comparison, and for more discrete perturbations, one can easily turn down the $\varepsilon$ radius to ensure more high fidelity user images.

Table 4: **Comparison of poisoning methods for facial recognition**. All poisons generated with $\varepsilon = 8/255$.

| POISON METHOD | AVERAGE ACCURACY (%,↓) |
|---|---|
| CLEAN ACCURACY | 86.00 |
| UNLEARNABLE EXAMPLES [HUANG ET AL., 2021] | 16.00 |
| OURS (CT) | **8.00** |

## 3.7 Less Data

So far, our comparisons to previous methods have been in the setting of full dataset poisoning, as was tested in Shen et al. [2019], Feng et al. [2019b], Huang et al. [2021], Fowl et al. [2021]. This setting is relevant to practitioners like social media companies who wish to modify all user data to prevent others from using data scraped from their platforms.

However, even in this setting, it may be the case that an actor scraping data has access to an amount of unperturbed data - either through previous scraping before a poisoning method was employed, or data scraped from another source. A concern that follows is how this mixture affects model training. Thus, we test the effectiveness of our poisons when different proportions of clean and perturbed data are used to train the victim model. Poisons are then considered effective if their use does not significantly increase performance over training on the clean data alone. In Tables 5, 6 we find that, poisoned data does not significantly improve results over training on only the clean data and often degrades results below what one would achieve using only the clean data. This is in comparison to similar experiments in Huang et al. [2021] where poisoned data was observed to be slightly helpful when combined with clean data in training a model.

Table 5: Effects of adjusting the amount of clean data included in the poisoned CIFAR-10.

| POISON METHOD\CLEAN PROPORTION | 0.1 | 0.2 | 0.5 | 0.8 |
|---|---|---|---|---|
| NONE (ONLY CLEAN DATA) | $82.30 \pm 0.08$ | $87.90 \pm 0.04$ | $92.48 \pm 0.07$ | $93.90 \pm 0.06$ |
| $\varepsilon = 8/255$ | $85.34 \pm 0.09$ | $88.23 \pm 0.09$ | $92.20 \pm 0.08$ | $93.65 \pm 0.07$ |

Table 6: Effects of adjusting the amount of clean data included in the poisoned ImageNet.

| POISON METHOD\CLEAN PROPORTION | 0.1 | 0.2 | 0.5 | 0.8 |
|---|---|---|---|---|
| NONE (ONLY CLEAN DATA) | 45.18 | 53.48 | 62.79 | 64.65 |
| $\varepsilon = 8/255$ | 47.81 | 54.24 | 61.20 | 63.95 |

## 3.8 Defenses

We have demonstrated the effects of our poisons in settings where the victim trains "normally". However, there have been several defenses proposed against poisoning attacks that a victim could potentially utilize. Thus, in this section, we test the effectiveness of several popular defenses against our method.

**Adversarial training**: Because our poisons are constrained in an $\ell_\infty$ ball around the clean inputs, and because we find that clean inputs are themselves adversarial examples for networks trained on poisoned data (more on this in the next section), it is possible that adversarial training could "correct" the poisoned network's behavior on the clean distribution. This is hypothesized in Tao et al. [2021], where it is argued that adversarial training can be an effective defense against delusive poisoning. We find in Table 7 that this is indeed the case. In fact, it is known that adversarial training effectively mitigates the success of several previous availability attacks including Huang et al. [2021], Feng et al. [2019b], Fowl et al. [2021]. However, it is worth noting that this might not be an ideal solution for the victim as adversarial training is expensive computationally, and it degrades natural accuracy to a level well below that of standard training. For example, on a large scale dataset like ImageNet, adversarial training can result in a significant drop in validation

accuracy. With a ResNet-50, adversarial training results in validation accuracy dropping from 76.13% to 47.91% - a drop that would be further exacerbated when adversarial training on poisoned data (cf. `https://github.com/MadryLab/robustness`).

**Data Augmentation**: Because our poisoning method relies upon slight perturbations to training data in order to degrade victim performance, it is conceivable that further modification of data during training could counteract the effects of the perturbations. This has recently been explored in Borgnia et al. [2021] and Geiping et al. [2021]. We test several data augmentations *not* known to the poisoner during crafting, although adaptive attacks have been shown to be effective against some augmentations [Geiping et al., 2021]. Our augmentations range from straightforward random additive noise (of the same $\varepsilon$-bound as poisons) to Gaussian smoothing. We also include popular training augmentations such as Mixup, which mixes inputs and input labels during training [Zhang et al., 2017], Cutmix [Yun et al., 2019], which mixes patches of one image with another, and Cutout [Cubuk et al., 2019] which excises certain parts of training images.

**DPSGD**: Differentially Private SGD (DPSGD) [Abadi et al., 2016] was originally developed as a differential privacy tool for deep networks which aims to inure a dataset to small changes in training data, which could make it an effective antidote to data poisoning. This defense adds noise to training gradients, and clips them. It was demonstrated to be successful in defending against targeted poisoning in some settings [Hong et al., 2020, Geiping et al., 2020]. However, this defense often results in heavily degraded accuracy. We test this defense with a clipping parameter $1.0$ and noise parameter $0.005$.

Table 7: **Effects of defenses against adversarial poisons**. All results averaged over 5 runs of a ResNet-18 victim model trained on class targeted, $\varepsilon = 8/255$ poisons. Although adversarial training improves results, none of these effectively recover the accuracy of a model trained on clean data.

| DEFENSE | VALIDATION ACCURACY (%) |
|---|---|
| BASELINE (CLEAN) | 94.56 |
| ADV. TRAINING | 83.01 |
| GAUSSIAN SMOOTHING | 11.94 |
| RANDOM NOISE | 6.55 |
| MIXUP | 15.86 |
| CUTMIX | 10.09 |
| CUTOUT | 8.11 |
| DPSGD | 24.61 |

Other than adversarial training, which has drawbacks discussed above, we find that previous proposed defenses are ineffective against our adversarial poisons, and can even degrade victim performance below the level of a model trained without any defense.

## 4 Analysis

Why do adversarial examples make such potent poisons? In Section 1, we motivate the effectiveness of adversarial poisons with the explanation that the perturbed data contains semantically useful information, but for the wrong class. For instance, an adversarially perturbed "dog" might be labeled as a "cat" by the crafting network because the perturbations contain discriminatory features useful for the "cat" class. In Ilyas et al. [2019], the authors discover that there exist image features which are both brittle under adversarial perturbations and useful for classification. Ilyas et al. [2019] explores these implications for creating robust models, as well as "mislabeled" datasets which produce surprisingly good natural accuracy. We investigate the implications that the existence of non-robust features have on data-poisoning attacks. We hypothesize that adversarial examples might poison networks so effectively because, for example, they teach the network to associate "cat" features found in perturbed data with the label "dog" of the original, unperturbed sample. Then, when the network tries to classify clean test-time data, it leverages the "mislabeled" features found in the perturbed data and displays low test-time accuracy. We confirm this behavior by evaluating *how* data is misclassified at test-time. We find that the distribution of predictions on *clean* data closely mimics the distributions of labels assigned by the network used for crafting after adversarial attacks, even though these patterns are generated from different networks.

To tease apart these effects, we conduct several experiments. First, we verify that the victim network does indeed train - i.e. reach a region of low loss - on the adversarial examples. This is in contrast to the motivation of [Shen et al., 2019, Huang et al., 2021] which try to *prevent* the network from training on poisoned data. We find that the victim network is able to almost perfectly fit the adversarial data. However, the accuracy of the victim network on the original, unperturbed training data is just as low as accuracy on the clean validation data, revealing an interesting duality - clean training data are adversarial examples for networks trained on their perturbed counterparts (see Table 8). But are adversarial examples simply so different from clean examples that learning on one is useless for performing inference on the other? Or do adversarial examples contain useful features but for the *wrong* class?

To investigate these hypotheses, we train models on the adversarial poisons crafted with the class targeted objective found in Equation 4 so that every image from a given class is perturbed into the same target class. We then observe the classification patterns of the victim network and compare these to the attack patterns of the crafting network. We find that poisoned networks confuse exactly the same classes as the crafting network. Specifically, a network trained on dog images perturbed into the cat class will then misclassify clean cat images as dogs at test time as the network learns to associate "clean" cat features found in the perturbations to the training dog images with the label dog. This behavior is illustrated in Figure 2.

To further confirm our hypothesis, we employ the re-labeling trick introduced in Ilyas et al. [2019], and we train the victim network on the "incorrect" labels assigned to the poisons by the *crafting* network. For example, if a dog image is perturbed into the "cat" class by an adversarial attack on the crafting network, we train on the perturbed dog image but assign it the "cat" label. We find that this simple re-labeling trick boosts validation accuracy significantly, for example from 6.25% to 75.69% on a victim ResNet-18 model (see Appendix Table 14). This confirms the finding of Ilyas et al. [2019] concerning non-robust features. One might think that this observation is useful for defending against adversarial poisons. However, in order to perform this label correction, the victim must have access to the crafting model which requires the victim to already possess the original non-poisoned dataset.

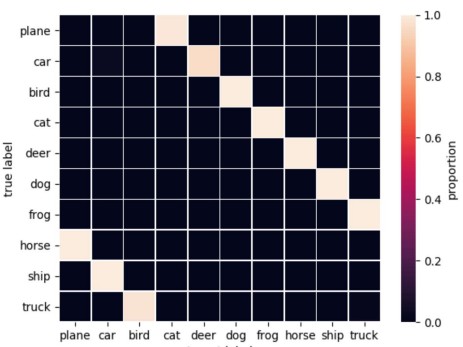 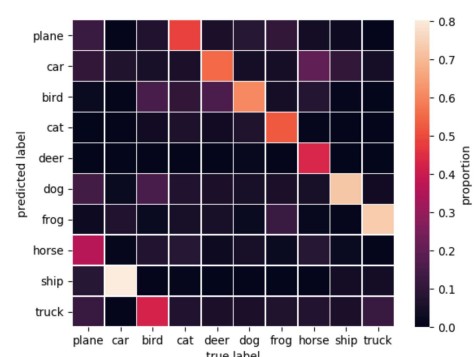

(a) Adversarial attack predictions of network used to craft adversarial poisons.

(b) Test-time predictions of network trained on adversarial poisons.

Figure 2: Classification heatmaps. **Left** - heatmap of predictions after the adversarial attack on the network used for crafting. **Right** - heatmap of clean test predictions after training a new network on adversarial poisons.

One further intricacy remains; even if adversarial examples contained no useful features, they may still encode decision boundary information that accounts for the increased validation accuracy - behaviour that has previously been demonstrated with out-of-distribution data in Nayak et al. [2019]. However, we find that training on data from a drastically different distribution (SVHN), labeled with the CIFAR-10 crafting network's predictions, fails to achieve comparable CIFAR-10 validation accuracy. This confirms that the adversarial CIFAR-10 images contain useful features for the CIFAR-10 distribution but are simply mislabeled.

Table 8: **Testing the victim on clean vs. adversarial training images.** Poisons are crafted on a CIFAR-10 trained ResNet-18 with 250 steps of PGD and differentiable data augmentation.

| MEASUREMENT \ VICTIM | VGG19 | RESNET-18 | GOOGLENET | DENSENET-121 | MOBILENETV2 |
|---|---|---|---|---|---|
| TRAINING ACC. ON POISONS | $99.95 \pm 0.00$ | $99.99 \pm 0.00$ | $99.99 \pm 0.00$ | $99.99 \pm 0.00$ | $99.94 \pm 0.00$ |
| ACC. ON CLEAN TRAIN DATA | $10.98 \pm 0.32$ | $6.16 \pm 0.16$ | $6.80 \pm 0.08$ | $7.06 \pm 0.13$ | $6.15 \pm 0.17$ |

## 5  Limitations

While our adversarial poisoning method does outperform current art on availability poisoning attacks, there are limitations to our method that warrant future investigation. We identify two primary areas of future research. First is the requirement of a clean trained model to generate adversarial examples, i.e. the $\theta^*$ parameter vector at which we craft poisons in Equation 3. This may be a reasonable assumption for practitioners like social media companies, but it does make the attack less general purpose. However, this is also a constraint with previous poisoning methods which require access to all the data being perturbed [Huang et al., 2011], or clean data to train an autoencoder [Feng et al., 2019b], or a large amount of clean data to estimate an adversarial target gradient [Fowl et al., 2021]. It has been shown that adversarial examples often transfer across initialization, or even dataset, so it is not inconceivable that this limitation of our method could be overcome. The second limitation is the success of adversarial training as a defense against delusive poisoning. This appears to be the trend across all availability attacks against deep networks to date. This defense is not optimal though as it leads to a severe drop in natural accuracy, and can be computationally expensive.

## 6  Conclusions

There is a rising interest in availability attacks against deep networks due to their potential use in protecting publicly released user data. Several techniques have been introduced leveraging heuristics such as loss minimization, and auto-encoder based noise. However, we find that adversarial attacks against a fixed network are more potent availability poisons than existing methods, often degrading accuracy below random guess levels. We study the effects of these poisons in multiple settings and analyze why these perturbations make such effective poisons. Our observations confirm a fundamental property of adversarial examples; they contain discriminatory features but simply for the wrong class. Our class-targeted attack leverages this property to effectively poison models on a variety of datasets.

## 7  Acknowledgements

This work was supported by DARPA GARD, the Office and Naval Research, and the National Science Foundation Division of Mathematical Sciences. Addition support was provided by the Sloan Foundation, JP Morgan Chase, and Capital One Bank.

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
