# A Appendix

## A.1 Training/Crafting details

Code for this project can be found at: `https://github.com/lhfowl/adversarial_poisons`. Below are descriptions of experimental settings and hyperparameters. Unless otherwise stated, for CIFAR-10, we train the crafting network for 40 epochs before generating the attacks. For testing, we use two primary setups. First, for the CIFAR-10 comparison and baseline Tables (1, 2), to be as fair and objective as possible, we use a totally third party testing setup found in the popular repo `https://github.com/kuangliu/pytorch-cifar`. Note for these runs, we updated our crafting procedure to use a slightly smaller step size ($0.05 \cdot \frac{8}{255}$).

For other CIFAR-10 ablations, for efficiency, we train victim models for 100 epochs with 3 learning rate drops with SGD optimizer. We craft with a step size of $0.1 \cdot \frac{8}{255}$. Unless otherwise stated, for CIFAR-10 experiments, we use 8 restarts on poison crafting, and perturbation pixels are initialized from $\mathcal{N}(0, \varepsilon^2)$.

For ImageNet, for efficiency, we use a pretrained crafting network (ResNet18 unless otherwise stated) taken from `https://pytorch.org/vision/stable/models.html` to craft the poisons. For memory constraints, we craft in batches of $25,000$, but note that this has no effect on the perturbations since the crafting is independent. Unless otherwise specified, we then train a randomly initialized model for 100 epochs with SGD using standard ImageNet preprocessing (resizing, center crops, normalization) with three learning rate drops. For the ablation studies on smaller proportions of data, for efficiency, we only train the ImageNet models for 40 epochs.

For crafting class targeted attacks, we select a random permutation of the labels. For CIFAR-10, this amounted to label $i \rightarrow i + 3$, and for ImageNet, we simply chose $i \rightarrow i + 3$.

For the facial recognition, we conduct our experiments following the partially unlearnable setting in Huang et al. [2021]. In this setting, we only assume ability to modify a subset of the face images, specifically, the faces images of the users that we want to protect. We first randomly split the Webface dataset into 80% training data and 20% testing data, and then randomly select 50 identities to be the users who want to hide their identities. We use the pretrained Webface model from Yi et al. [2014] to generate the class targeted adversarial examples with $\varepsilon = 8/255$. We then combine the protected data together with the the remaining 10525 identities to form the new training dataset. We then train an Inception-ResNet following the standard procedures in Taigman et al. [2014]. We visualize perturbations in Appendix Figure 5.

### A.1.1 Hardware and time considerations

We run our experiments on a heterogeneous mixture of resources including Nvidia GeForce RTX 2080 Ti GPUs, as well as Nvidia Quadro GV100 GPUs. Crafting and training time vary widely depending on the dataset and hyperparameter choices (restarts). However, a typical crafting experiment for CIFAR-10 will take roughly 6 hours to train and craft with 4 2080 Ti GPUs. This can be reduced by a factor of roughly 8 if one utilizes pretrained models, and only performs 1 restart during poison creation.

## A.2 Visualization

In Figure 3, we visualize randomly selected perturbed images at different $\varepsilon$ levels. As with adversarial attacks, and other poisoning attacks, there is a trade-off between visual similarity and potency of the perturbations.

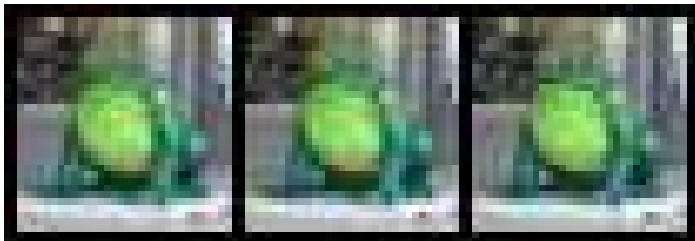

Figure 3: Randomly selected example perturbations to CIFAR-10 datapoint (class "frog"). **Left**: unaltered base image. **Middle**: $\varepsilon = 4/255$ perturbation. **Right**: $\varepsilon = 8/255$ perturbation. Networks trained on perturbations including the one on the right achieve below random accuracy.

In Figure 5, we visualize perturbed identities from the Webface dataset.

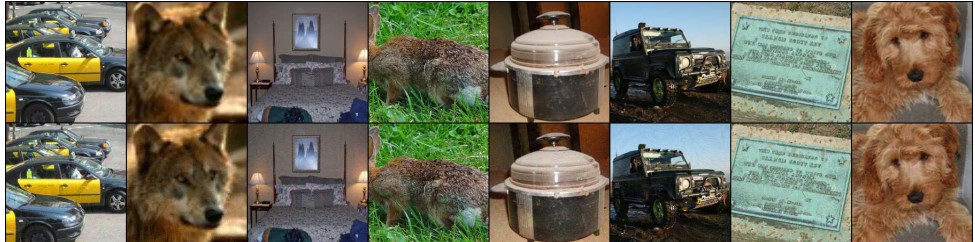

Figure 4: Randomly selected example perturbations to ImageNet datapoints. Perturbations bounded by $\varepsilon = 8/255$.

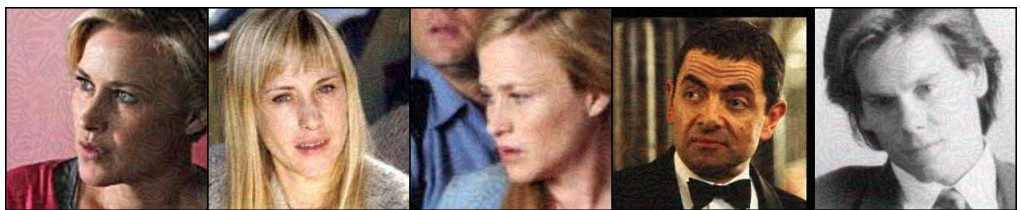

Figure 5: Samples of poisoned Webface images $\varepsilon = 8/255$

## A.3 Adversary comparison

We find that a PGD based attack supersedes other common adversarial attacks in poison efficiency - a behavior that has been observed in classical adversarial attacks as well. These results can be found in Table 9.

Table 9: **Validation accuracies of victim models trained on data generated by different adversarial attacks.** Tested in the black-box setting on randomly initialized models on CIFAR-10.

| METHOD \ VICTIM | VGG19 | RESNET-18 | GOOGLENET | DENSENET-121 | MOBILENETV2 |
|---|---|---|---|---|---|
| PGD W/ AUG | $10.98 \pm 0.27$ | $6.25 \pm 0.17$ | $7.03 \pm 0.12$ | $7.16 \pm 0.16$ | $6.11 \pm 0.17$ |
| CW | $59.82 \pm 1.36$ | $48.40 \pm 3.24$ | $19.25 \pm 1.15$ | $43.40 \pm 2.76$ | $45.62 \pm 4.87$ |
| FGSM | $65.32 \pm 0.58$ | $76.35 \pm 0.08$ | $47.50 \pm 1.06$ | $59.44 \pm 1.28$ | $74.77 \pm 0.43$ |
| FEATURE EXPLOSION | $82.41 \pm 0.38$ | $83.26 \pm 0.94$ | $78.53 \pm 0.49$ | $81.83 \pm 0.46$ | $82.30 \pm 0.88$ |

## A.4 Crafting Ablations

In Table 10, we find that the more steps we perform in the PGD optimization of adversarial poisons, the more effective they become. However, the poisons still degrade validation accuracy at lower numbers of steps.

Table 10: **Comparison of different number of crafting steps for our adversarial poisoning method.** All poisons generated by ResNet-18 with steps of PGD, with differentiable data augmentation, and class-targeted objective.

| STEPS \ VICTIM | VGG19 | RESNET-18 | GOOGLENET | DENSENET-121 | MOBILENETV2 |
|---|---|---|---|---|---|
| 50 STEPS | $25.10 \pm 0.60$ | $16.53 \pm 0.26$ | $18.51 \pm 0.34$ | $18.91 \pm 0.34$ | $15.29 \pm 0.40$ |
| 100 STEPS | $16.06 \pm 0.41$ | $9.87 \pm 0.26$ | $11.66 \pm 0.33$ | $13.42 \pm 0.28$ | $10.38 \pm 0.22$ |
| 250 STEPS | $10.98 \pm 0.27$ | $6.25 \pm 0.17$ | $7.03 \pm 0.12$ | $7.16 \pm 0.16$ | $6.11 \pm 0.17$ |

## A.5 Network Variation

Here we test how perturbations crafted using one network transfer to other networks. While it has been demonstrated that adversarial examples used as evasion attacks can often transfer across architectures, we find that adversarial examples as poisons also transfer across different architectures in Table 12.

In addition to experimenting with different crafting network architectures, we also vary other factors of the crafting network. For example, we experiment how using an adversarially robust crafting network affects

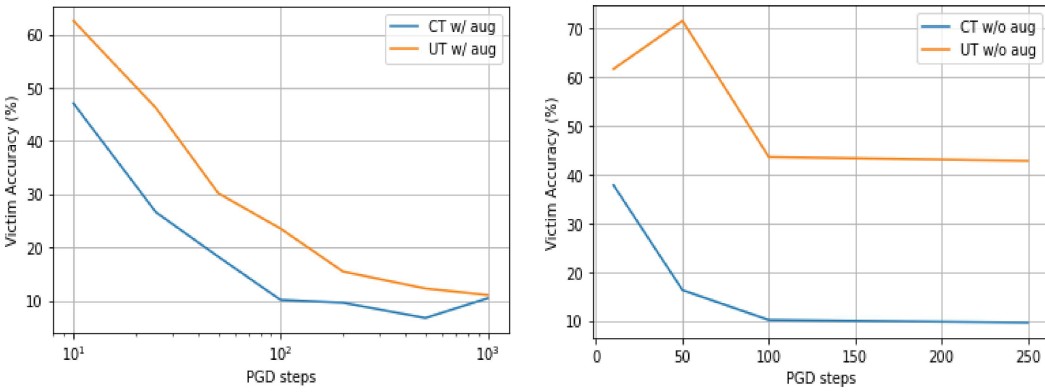

(a) Ablating the number of PGD steps used for crafting. Step size is fixed at $0.05 \cdot \frac{8}{255}$

(b) Ablating the number of PGD steps used for crafting, without augmentation. Step size is fixed at $0.05 \cdot \frac{8}{255}$

Figure 6: Victim network accuracy as a function of PGD steps with and without augmentation. Augmentation appears necessary to produce powerful untargeted poisons.

Table 11: **A comparison of different optimizers, crafting objectives, and use of differentiable data augmentation in crafting.** All poisons crafted with bound $\varepsilon = 8/255$. If not otherwise stated, poisons are crafted with PGD, differentiable data augmentation, and our class-targeted objective.

| METHOD \VICTIM | VGG19 | RESNET-18 | GOOGLENET | DENSENET-121 | MOBILENETV2 |
|---|---|---|---|---|---|
| PGD W/ AUG | $10.98 \pm 0.27$ | $6.25 \pm 0.17$ | $7.03 \pm 0.12$ | $7.16 \pm 0.16$ | $6.11 \pm 0.17$ |
| SIGNADAM W/ AUG | $8.92 \pm 0.27$ | $6.17 \pm 0.27$ | $6.75 \pm 0.18$ | $6.42 \pm 0.23$ | $7.15 \pm 0.22$ |
| SIGNADAM W/O AUG | $9.96 \pm 0.22$ | $6.96 \pm 0.15$ | $7.41 \pm 0.15$ | $7.84 \pm 0.29$ | $8.22 \pm 0.23$ |
| CW LOSS | $59.82 \pm 1.36$ | $48.40 \pm 3.24$ | $19.25 \pm 1.15$ | $43.40 \pm 2.76$ | $45.62 \pm 4.87$ |
| FEATURE EXPLOSION LOSS | $82.41 \pm 0.38$ | $83.26 \pm 0.94$ | $78.53 \pm 0.49$ | $81.83 \pm 0.46$ | $82.30 \pm 0.88$ |

Table 12: **Results varying the *crafting* network for the poisons.** All poisons crafted with bound $\varepsilon = 8/255$, PGD, differentiable data augmentation, class-targeted objective.

| CRAFTING \TESTING | VGG19 | RESNET-18 | GOOGLENET | DENSENET-121 | MOBILENETV2 |
|---|---|---|---|---|---|
| RESNET-18 | $10.98 \pm 0.27$ | $6.25 \pm 0.17$ | $7.03 \pm 0.12$ | $7.16 \pm 0.16$ | $6.11 \pm 0.17$ |
| RESNET-50 | $17.86 \pm 0.57$ | $9.71 \pm 0.12$ | $11.44 \pm 0.21$ | $10.64 \pm 0.46$ | $6.82 \pm 0.18$ |
| VGG19 | $20.88 \pm 0.84$ | $18.53 \pm 1.15$ | $21.48 \pm 0.67$ | $23.77 \pm 0.74$ | $17.59 \pm 0.56$ |
| MOBILENETV2 | $21.66 \pm 0.26$ | $12.42 \pm 0.17$ | $14.84 \pm 0.29$ | $15.95 \pm 0.18$ | $9.60 \pm 0.24$ |
| CONVNET | $15.43 \pm 0.34$ | $10.05 \pm 0.20$ | $11.88 \pm 0.17$ | $10.30 \pm 0.11$ | $7.95 \pm 0.26$ |

performance of the poisons. We find that adversarially trained models produce ineffective poisons. This is in line with the findings of [Ilyas et al., 2019] that robust models leverage a different set of discriminatory features - ones less brittle to perturbation - during classification. These results can be found in Table 13.

Table 13: Poison generation using a robust crafting model.

| CRAFT MODEL \VICT. MODEL | VGG19 | RESNET-18 | GOOGLENET | DENSENET-121 | MOBILENETV2 |
|---|---|---|---|---|---|
| ROBUST RESNET-18 | $85.58 \pm 0.05$ | $81.79 \pm 0.57$ | $80.99 \pm 0.17$ | $80.39 \pm 0.27$ | $77.70 \pm 1.27$ |

## A.6 Relabeling Trick

## A.7 Learning Without Seeing

We have seen that it is not necessary to have "correctly" labeled data (labeled with ground truth labels) in order for a network to achieve good performance at test-time. We can extend this to the question: does one need ground truth data *at all* to learn how to classify? For example, can a network learn how to classify cats without ever seeing an image of a cat, but instead only seeing images of dogs perturbed to look like cats? We find the

Table 14: **Training the victim with labels corrected to the "adversarial labels".** Poisons are crafted on a CIFAR-10 trained ResNet-18 with 250 steps of PGD and differentiable data augmentation.

| DATA \ VICTIM | VGG19 | RESNET-18 | GOOGLENET | DENSENET-121 | MOBILENETV2 |
|---|---|---|---|---|---|
| UNCORRECTED CIFAR-10 | $10.98 \pm 0.27$ | $6.25 \pm 0.17$ | $7.03 \pm 0.12$ | $7.16 \pm 0.16$ | $6.11 \pm 0.17$ |
| CORRECTED ONE-HOT CIFAR-10 | $74.95 \pm 0.31$ | $78.98 \pm 0.25$ | $77.72 \pm 0.37$ | $78.55 \pm 0.36$ | $74.33 \pm 0.24$ |
| CORRECTED SOFTMAX CIFAR-10 | $75.88 \pm 0.25$ | $75.69 \pm 0.25$ | $73.57 \pm 0.47$ | $70.26 \pm 0.32$ | $69.46 \pm 0.32$ |
| CORRECTED ONE-HOT SVHN | $31.13 \pm 0.49$ | $30.19 \pm 0.16$ | $40.71 \pm 0.12$ | $40.31 \pm 0.43$ | $34.18 \pm 0.29$ |

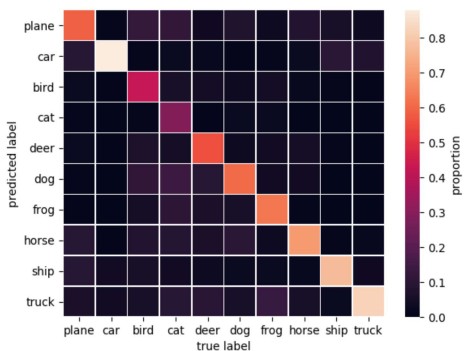

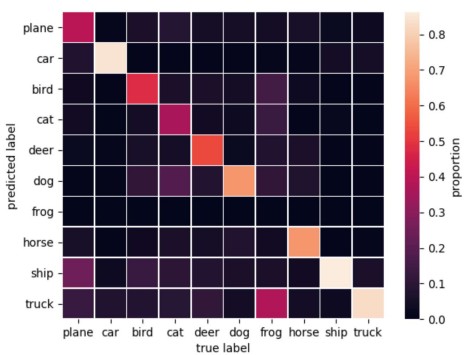

(a) Test-time predictions of network trained on label-corrected, class-targeted poisons.

(b) Test-time predictions of network trained on label-corrected, class-targeted poisons without any images of "cats".

Figure 7: Classification heatmaps. **Left** - heatmap of clean test predictions from network trained on label-corrected, class based targeted poisons. **Right** - heatmap of clean test predictions from network trained on same poisons, without any "cat" images.

answer is yes. Specifically, we conduct an experiment where we craft targeted, class-based poisons (i.e. all images of one class are perturbed via targeted attacks into another class). We then train on the label-corrected full data, and also train on a label-corrected pruned training set without any images from the cat class. Interestingly, we find that the network trained on the ablated set is able to classify clean, test-time images of cats having never seen an example during training! Moreover, the network fails to classify clean images from the class into which cats were perturbed (class 6, "frog") under the targeted attack. Instead, clean test-time frogs are more likely to be classified as class 9 ("truck") - the class where frog images were perturbed into under the crafting attack. Interestingly, this demonstrates that the network not only learns to associate the perturbed features of frogs with the label "truck", but also features of clean frog images even though the network never encountered "clean" frog features during training because the cat class was ablated. These results can be found in Figure 7.

## A.8   ImageNet Comparison

While our main ImageNet results are in a realistic setting of training for 100 epochs, previous art has poisoned ImageNet victims trained for 40 epochs. We compare our method to their's here and find that our clas targeted attack far outperforms their attack based on gradient alignment.

Table 15: Effects of adjusting the amount of clean data included in the poisoned ImageNet. The same number of gradient updates were used for each model (equivalent to 40 epochs for full sized ImageNet).

| POISON METHOD \ POISON BOUND | $\varepsilon = 8/255$ |
|---|---|
| CLEAN | 65.70 |
| ALIGNMENT | 37.58 |
| OURS | **1.57** |

## A.9 Instability of Untargeted Attacks

While untargeted adversarial poisons are able to significantly degrade the validation accuracy of a victim models, it is a weaker attack than our class targeted attack, and it can be more brittle to poison initialization. Some poisons generated with untargeted attacks achieve much worse performance than others. We hypothesize this is because the poison initialization has a large effect on which class the untargeted attack is perturbed into, thus affecting the feature confusion aspect to poisoning. As mentioned in the main body, we perform modest hyperparameter searching (poison initialization) for the untargeted CIFAR-10 attack. Specifically, for comparison to other methods, for the untargeted attack, we seed the poisons and model initialization for poison crafting, and take the best run from a modest number of trials ($< 15$). This is not unreasonable for a practitioner like a social media company as they potentially have access to the entire dataset which they intend to poison, and can choose the best set of perturbations. In these trials, we observe that the average success (%) of these poisons (now varying over initialization of the poisons) is $19.17$ with standard deviation $2.87$. The most potent poisons in these runs degrade victim accuracy to $11.94\%$, and it is on these poisons that we run comparisons. Note that these do not represent the *most* potent untargeted poisons we have found, but rather a "reasonable" best set a poisoner can hope to find. Even more potent poisons can be found linked through our Github `https://github.com/lhfowl/adversarial_poisons`.

On the other hand, our class-targeted objective is much more stable to poison/model initialization, and thus we do not perform hyperparameter searches for these runs. The relative weakness of the untargeted objective is evident in our ImageNet results in Table 3. We hypothesize the success of the untargeted variant depends on how "well distributed" the attack matrix is (which in turn depends on poison initialization). We have included a heatmap of an untargeted attack in Figure 8.

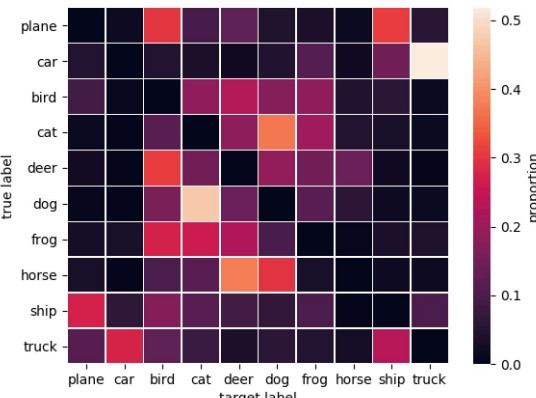

Figure 8: An example heatmap of an untargeted poisoning measuring the attack distribution on the crafting network. Note how this attack "well distributes" the target labels (i.e. not every class is perturbed into the same class). We hypothesize this is important for a successful attack.