# OpenReview forum: "Adversarial Examples Make Strong Poisons"
_NeurIPS.cc/2021/Conference — NeurIPS 2021 Poster_

### Official Review · Reviewer_WUSL · 2021-07-13

**Rating:** 6
**Confidence:** 2

**Summary:**

This paper proposes a data poisoning approach based on the adversarial attack technique. Specifically, to generate the perturbed samples, the poisoning model aims at maximizing the traditional cross-entropy loss or minimizing the loss with the wrong labels. Adversarial example generation approach PGD is adopted to generate the poisons. Experiments on several datasets show that training on the perturbed samples would significantly reduce the performance of various deep learning models.

**Limitations And Societal Impact:**

As mentioned in the paper, the proposed approach assumes the knowledge of a pre-trained model to generate the perturbation, which makes the approach not very practical in the real world. Many attack strategies, such as backdoor attacks, usually assume to inject limited instances to training samples to damage the model.

**Main Review:**

The main contribution of this work is to leverage the idea of adversarial example generation for data poisoning. The perturbation generation is directly based on the existing approaches. The authors also conduct substantial experiments to show the effectiveness of the proposed approach.

**Time Spent Reviewing:**

4

---

> ### Author Response · Authors · 2021-08-10
> **Response to Reviewer WUSL**
>
> Thank you for your time spent reviewing and thoughtful feedback. Below are our responses to your concerns.
>
> &nbsp;
>
> ### On limitations
>
> &nbsp;
>
> > the proposed approach assumes the knowledge of a pre-trained model to generate the perturbation, which makes the approach not very practical in the real world ...
>
> We emphasize that our attack type is completely black-box and does not assume the organization seeking to protect their data has any knowledge of the “victim” (who seeks to scrape the data) or their architecture; the attack is generated on a surrogate model.  For our unique threat model, it is entirely realistic for the poisoner (e.g., a social media company seeking to protect their data) to create a surrogate model since they have all the original clean data on which to train the surrogate model.  Moreover, in Table 1, we show that it is not necessary for the poisoner’s surrogate model to resemble the victim’s model (in fact, we observed our best poisoning performance in situations where the models are different).
>
> Your comment does bring up an interesting experiment, though. Suppose a company receives user data in a "stream" of uploads, rather than all at once.  In this case, there may not be a large corpus of data that can be used to produce a pre-trained model.  How does this affect the potency of the generated poisons? To this end, we have since run an experiment wherein the poisoner trains a surrogate model on only a *small* amount of data (relative to the data they perturb). In this case, our method still produces potent poisons. For example, using a crafting model trained only on 10% of the overall clean data, our class-targeted poisons still degrade a victim ResNet-18's accuracy to 12.36% +/- 0.37%. We have since added this to our manuscript, and we thank you for your comment.

---

### Official Review · Reviewer_7LBn · 2021-07-15

**Rating:** 6
**Confidence:** 4

**Summary:**

In this paper, the authors show that test-time adversarial examples, which are originally crafted to fool a well-trained model at inference time, can be used to construct strong availability attacks at training time. The resulting attacks achieve state-of-the-art results on multiple datasets and can successfully transfer across many settings, such as model initialization, architecture, learning rate scheduler, optimizer, etc.

**Limitations And Societal Impact:**

Although many limitations of the work are studied through experiments, I still have some suggestions, as listed below.

- It would be better if the authors could plot the data in Table 10 using curves, to show the trend of the decreasing accuracy as the number of PGD steps increases, with more fine-grained hyperparameters (e.g., plotting every 20 steps, up to 1000 steps). I'm curious about the limits of this poison.
- [2] and [3] showed that class-wise noises perform better than sample-wise noises, for both error-maximizing attacks and error-minimizing attacks. It would be better if this work can also consider the class-wise noises.
- Section 4 of the manuscript (and Figure 2) only explains why the targeted adversarial examples (i.e., Equation 4) work for poisoning. However, [1] provided a more concise explanation through permutation-symmetry, which directly implies the use of targeted adversarial examples as poisoning attacks.
- It remains unclear why the untargeted adversarial examples (i.e., Equation 3) are also effective (sometimes more effective than targeted adversarial examples as shown in Table 1). Since the originality of this work is limited, it would be helpful to provide a novel and insightful analysis like this.
- I note that, in Table 2, the results of UNLEARNABLE EXAMPLES and DEEPCONFUSE are different from those reported in their original papers. Please clarify this.

**Main Review:**

#### **Originality**

My main concern about the paper is its originality. The idea, adversarial examples can make availability attacks, is not new to the community. It was first proposed by Nakkiran [1]. He simply predicted the success of the idea through permutation-symmetry of the labels, and showed its effectiveness with numerical results. For attribution in academic contexts, please cite his work [1] properly.

Furthermore, the exact idea was also experimented in previous works, such as the error-maximizing noise attack in Huang et al. [2] and the P1 attack in Tao et al. [3], just with slightly different optimization hyperparameters. These previous efforts about the idea should not be ignored in the main text.

#### **Significance**

Since the idea is not new, the main insight I can tell from the manuscript is that adversarial examples can make "strong" availability attacks. Contradicting to the results reported by Huang et al. [2], where they concluded that the error-maximizing attack is inferior to their proposed error-minimizing attack, this work shows that the same error-maximizing noise attack can actually surpass the error-minimizing attack (6.25% vs 19.85% in Table 2).

I note that one possible crux is the number of PGD steps used for crafting poisons: [2] adopted only 20 steps for the error-maximizing attack, while this work executes 250 steps. From Table 10 in Appendix, it turns out that the number of PGD steps is crucial for achieving high performance. Therefore, I suggest the authors to clarify this earlier in the main text. This is an important point that differentiates this work from previous works.

#### **Quality**

Main claims are experimentally supported. However, neither a new method nor theoretical analysis is proposed.

#### **Clarity**

The writing of this paper is mostly clear and straightforward. A minor concern is that Section 3 is too long. It would be better to separate Section 3 into two sections, one describing the problems and methods, the other showing the experiments.

Reference:

[1] Nakkiran, https://distill.pub/2019/advex-bugs-discussion/response-5/

[2] Huang et al., https://arxiv.org/abs/2101.04898

[3] Tao et al., https://arxiv.org/abs/2102.04716


-----
After rebuttal:

I am glad to see that my suggestions helped to identify the contributing factors for attack effectiveness. It turns out that three strong factors emerge during the rebuttal:
- the variance of the attack effectiveness itself
- the use of differentiable data augmentation
- the number of PGD steps

I would like to suggest the authors to reflect these key factors (and other minor factors) more clearly in the main body. Surely, achieving SOTA performance itself is a good contribution, but for this venue, it alone may not be enough. In my opinion, another contribution of this work would be to sufficiently clarify why prior implementations of adversarial examples cannot achieve SOTA. Thus, when combining the two contributions, I would prefer this paper.

Overall, I am leaning to accept this paper on the condition that the authors can further clarify the confusion about the variance, the use of differentiable data augmentation, and the motivation for the choice of the number of PGD steps.

**Time Spent Reviewing:**

9

---

> ### Author Response · Authors · 2021-08-10
> **Response to Reviewer 7LBn**
>
> Thank you for your time, and thoughtful and knowledgeable comments on our work. Below we include responses to your concerns.
>
> &nbsp;
>
> ### On originality
>
>
> We have since added a reference to the Distill blog post [1]. Thank you for bringing this to our attention. We agree that the idea does follow nicely from the work of Ilyas et al., which we cited in our original submission.
>
> We develop this approach as a practical poisoning attack, including extensive experimentation and analysis. Furthermore, the topic of secure dataset release has been growing in popularity (e.g. Unlearnable Examples, DeepConfuse, Gradient Alignment, and a recent ICML workshop paper: "Disrupting Model Training with Adversarial Shortcuts"). Relative to these existing methods, our approach is currently the state of the art.  Because our method roundly beats all previous methods, we believe these results warrant publication for the community (provided prior works are properly cited, of course).
>
> Regarding [3], our method was developed prior to the appearance of [3] on arXiv (between ICML and NeurIPS deadlines), and we consider it concurrent work. While they do suggest our class targeted attack, the main focus of [3] is as a defense. They do not perform extensive experimentation/analysis on this topic, and they do not seem to experiment with our untargeted objective. We do cite this paper in the defense section, and we have since changed Section 3.2 in our manuscript to reflect that this work concurrently suggests a class targeted attack.
>
> As for the difference in effectiveness relative experiments on [2] (Unlearnable Examples), we have included our evaluation code in the supplemental materials, and we have additionally tested our poisoned dataset in a widely used popular CIFAR-10 training [repo](https://github.com/kuangliu/pytorch-cifar) and found similar results in its setup (8.15% accuracy on a ResNet18 trained on our untargeted attack).  We have several hypotheses for the difference in attack success (more on this below w.r.t. your comments on PGD steps, untargeted attacks). It is also worth noting that [2] does not experiment with class-targeted error maximization, but rather a *class-wise* error maximization which adds an identical pattern to each image of a given class. As they note in their work, this should be easier for defenders to filter out.  Nonetheless, we have updated section 3.2 to reflect that an untargeted objective was considered in [2].
>
>
> &nbsp;
>
> ### On loss maximizing perturbations/untargeted attacks
> > I note that one possible crux is the number of PGD steps used for crafting poisons:
>
> You are correct that going from 50 steps of PGD to 250 does result in increased poison performance (16.53% to 6.25%). However, both are still lower than was quoted for the loss maximization numbers in [2], so we suspect there are other contributing factors arising from differences in the crafting routine. For example, we initialize our perturbations randomly, whereas it [2] initializes perturbations to zero (more on this below).  However, we agree that this should be clarified, so upon your recommendation, we have since included a more fine-grained study of PGD steps in the Appendix and mentioned the increased performance due to more steps in the main body.
>
> > It remains unclear why the untargeted adversarial examples (i.e., Equation 3) are also effective
>
> The untargeted results were intriguing to us in light of [2]. We confirm that this is not an artifact of our testing regime and have since generated their samplewise “max-max” attack, and tested it in our framework (more details in another response below). In our framework, their loss maximization attack only degrades a victim ResNet-18 to 87.21% validation accuracy.
>
> It is worth stating that we also observe that untargeted attacks are generally weaker, and we do note a steep dropoff (in comparison to class targeted attack) on ImageNet (see Table 3). We hypothesize that this is because of the more complicated label space of ImageNet. Unlike a problem with fewer classes, ImageNet scores are generally sprinkled over a large number of classes, and scores for the correct class may be relatively low (even if still larger than scores for other classes).  For this reason, an untargeted evasion attack on ImageNet can succeed simply by slightly lowering the score for the true class, and without introducing any strong class-specific features for any other class or raising the confidence in any other class.  See Section A9, where we discuss conditions wherein we observe successful untargeted attacks. We agree this is an important point and have since updated the main body to also include this detail, and expanded its discussion. Put simply, based on prediction heatmaps, we believe that the performance of untargeted attacks is more dependent on poison initialization and the crafting model, and successful untargeted poisons have a "well distributed" attack matrix (see Appendix Figure 6). That is to say the attack matrix mimics some permutation on the label space. We have made this more clear in the main body.
>
> In terms of the discrepancy on CIFAR-10, several differences in crafting, including perturbation initialization, combined with number of PGD steps, number of restarts, as well as our use of differentiable data augmentation, etc., likely lead to the difference. With the above explanation on the fragility of loss maximizing perturbations, a fixed, zero initialization especially (as seen in [2]’s [repo](https://github.com/HanxunH/Unlearnable-Examples) ) could lead to suboptimal perturbations. We have since made this more clear in the experimental section.
>
>
> &nbsp;
>
> ### On other concerns/comments
>
> > It would be better if the authors could plot the data in Table 10 using curves,
>
> Thank you for the suggestion. We have updated this Table with a more extensive, and fine-grained ablation on step number.
>
> &nbsp;
>
> > [2] and [3] showed that class-wise noises perform better than sample-wise noises, ... It would be better if this work can also consider the class-wise noises.
>
> We did not initially experiment with class-wise noise as this can be easier to detect and remove downstream (as noted in [2]). It appears as though class-wise noise works via a straightforward mechanism - “watermarking” all images of one class with a pattern the network learns to associate with that class label. In fact, according to [2], random class-wise noise actually performs better than optimized class-wise noise, making this a less interesting domain of research. This has also recently been studied in this [work](https://arxiv.org/abs/2106.06654). We do agree this does seem like a powerful mechanism for degrading accuracy though.
>
> &nbsp;
>
> > [1] provided a more concise explanation through permutation-symmetry
>
> While we think this is good intuition, and in fact true, the hypothesis was not empirically validated in [1]. Our label-correction and prediction heatmap experiments do back this up.  We have added further discussion and related our work to [1].
>
> &nbsp;
>
> > I note that, in Table 2, the results of UNLEARNABLE EXAMPLES and DEEPCONFUSE are different from those reported in their original papers
>
> Thank you for pointing out confusion over our testing regime. We have since clarified this in the main body. We generated poisons from the respective code repositories for these two methods, and then we test the generated poisons on a standard vanilla victim training routine from adapted with slight modifications from a 3rd party [repo](https://github.com/kuangliu/pytorch-cifar) in order to provide a consistent and realistic testing environment. Also, instead of loading perturbations as tensors, we add them to image data, and then re-save the image data to mimic the most realistic poisoning scenario. Finally, DeepConfuse originally used a slightly higher bound for their attack than our paper or Unlearnable Examples, so we adjust the epsilon bound of DeepConfuse accordingly, so that all attacks are consistent.

---

> > ### Comment · Reviewer_7LBn · 2021-08-20
> > **Thanks for your response**
> >
> > I appreciate the authors for the detailed response. My concern about originality has been somewhat addressed.
> >
> > My original concern about originality: Considering that the idea "adversarial examples are effective poisons" is not new in the literature, I thought it is important to make a proper positioning of this work. The original manuscript did not mention previous efforts on this idea, especially the work of Nakkiran [1], who was the first to propose this idea.
> >
> > I am glad that the authors frankly acknowledged this point in their rebuttal, and the authors also promised that they will reflect the previous efforts on the idea [1-3] in the main text. I believe that this reflection will help the positioning of this work (In light of this, I would like to increase my score).
> >
> > Having said that, I am still not sure whether the contribution of this work is solid enough to be accepted. The performance gap of "adversarial examples as poisons" between this work and the prior works [1] [2] [3] is large; however, it is still unclear which factor contributes to this. The authors themselves did not fully figure this out in the last response. The contributing factors may include perturbation initialization, the adoption of differentiable data augmentation, the number of PGD steps (probably the longer the better), and the number of training epochs of the surrogate model (probably the 40 epochs used in this work is better than full training used in [2] [3]). This work still lacks a complete picture of these factors. It would be very helpful if the authors could reveal the key crux underlying the strong performance of adversarial examples.
> >
> > The authors said in the last response that they have included a fine-grained study of PGD steps offline. Could you post the results here? It might be helpful. Thank you in advance.

---

> > > ### Author Response · Authors · 2021-08-22
> > > **Response (part 2)**
> > >
> > > We again thank the reviewer for their insightful comments and questions. We truly do appreciate your feedback and the time you are dedicating to a thorough review.
> > >
> > > We have since run several experiments aiming to bridge the gap in baselines between the findings of [2] (and to an extent [3]) with our work. While [2] achieves impressive results with their loss minimization objective, they argue that because adversarial training does not significantly degrade natural accuracy, a loss maximization objective would similarly produce weak poisons. However, because adversarial training continually updates the perturbed inputs, the optimization problem becomes very different from training a network on fixed loss maximization perturbations. We argue that a loss maximization objective can indeed produce powerful poisons. To this end, we find several factors not employed in the pipeline of [2] which make loss maximization an incredibly potent poisoning objective. Because of these differences, the loss maximization baselines in [2] seem deflated. Accordingly, we focus these new experiments in a similar setting to [2].
> > >
> > >
> > > * One strong factor is differentiable data augmentation. During crafting, we backprop through differentiable augmentations including translations, crops, and horizontal flips of poisons. This seems to have a large effect on the potency of untargeted poisons (Table 3).
> > >
> > > * We also find that decreasing the number of PGD steps does indeed degrade the potency of the poisons. For example, going from 250 steps to 20 steps increases the victim's validation accuracy by around 30%. Thus, the 20 steps used in [2] lead to suboptimal poisons. Note that the increased computation from more PGD steps is made up for by us not retraining the model during crafting as in [2] (see final paragraph). Thank you for suggesting this line of inquiry.
> > >
> > > * Crafting model training epochs has a mixed effect on poison potency (Tables 5,6). Although the general trend seems to be that at a lower number of PGD steps, more training epochs for the crafting model decreases poison potency (53.03% -> 33.42%). This likely is a factor in differences in results from [2] as they train their crafting model for more epochs than we do. However, the effects are less interpretable at a higher number of PGD steps.
> > >
> > > * Initializing poisons to zero (rather than randomly) seems to have mixed effects on poison potency (Table 2). At lower PGD steps, it seems to harm the potency noticeably (40.58% -> 50.21%). However, at larger step numbers, the results are not noticeable, and even go the other direction sometimes. We think this is less important than the factors listed above.
> > >
> > >
> > >
> > > We have adjusted both our appendix and our manuscript to make these findings clear to readers. We think the findings on untargeted results are especially interesting as a standard untargeted objective *can* be used to craft potent poisons with adjustments from previous findings ([2]) on this topic. Thank you for inspiring this line of experiments. We are still going through [2]'s code to find other sources of difference, but this is a lengthy process. Finally, we would like to stress once more that to the best of our knowledge, the practical attacks which we put forward achieves state of the art numbers on a field of research that is of considerable interest to the community, and is quickly heating up.
> > >
> > > #### Experimental Setup
> > > These tables were generated via our submitted code with only 1 restart, and 1 validation run for time constraints. We will reproduce these tables with our standard number of restarts and more validation runs to get error estimates. Also, to remove any concern about our testing environment, all of these numbers are generated *directly* from the popular 3rd party repo linked in our previous comment (as opposed to our included poison evaluation code which was modified from the 3rd party repo).  We are still going through [2]'s code to find other sources of difference, but this is a lengthy process. We are also currently running more ablations on factors such as PGD step size (we use a slightly smaller step size than [2]), restarts, etc., although we hypothesize these factors are comparatively less important. Nonetheless, we will post results as they come in. All poisons are crafted with a ResNet-18, and for consistency, we seeded the crafting model initialization.
> > >
> > > &nbsp;
> > >
> > > ##### Table 1:
> > > > ResNet-18 CIFAR-10 Test Accuracy under untargeted attack, varying the number of PGD steps (Baseline).
> > >
> > > | PGD Steps: | 20     | 30     | 40     | 50     |
> > > | :--------- | :----- | :----- | :----- | :----- |
> > > | Accuracy (%):  | 40\.58 | 30\.53 | 30\.13 | 29\.19 |
> > >
> > > &nbsp;
> > >
> > > ##### Table 2:
> > > > ResNet-18 CIFAR-10 Test Accuracy under untargeted attack, varying the number of PGD steps with poisons identically initialized to zero.
> > >
> > > | PGD Steps: | 20     | 30     | 40     | 50     |
> > > | :--------- | :----- | :----- | :----- | :----- |
> > > | Accuracy (%):  | 50\.21 | 35\.10 | 28\.86 | 26\.97 |
> > >
> > > &nbsp;
> > >
> > > ##### Table 3:
> > > > ResNet-18 CIFAR-10 Test Accuracy under untargeted attack, varying the number of PGD steps without differentiable data augmentation.
> > >
> > > | PGD Steps: | 20     | 30     | 40     | 50     |
> > > | :--------- | :----- | :----- | :----- | :----- |
> > > | Accuracy (%):  | 79\.25 | 53\.27 | 52\.63 | 51\.69 |
> > >
> > > &nbsp;
> > >
> > > ##### Table 4:
> > > > ResNet-18 CIFAR-10 Test Accuracy under targeted attack, varying the number of PGD steps without differentiable data augmentation.
> > >
> > > | PGD Steps: | 20     | 30     | 40     | 50     |
> > > | :--------- | :----- | :----- | :----- | :----- |
> > > | Accuracy (%):  | 25\.35 | 19\.96 | 17\.25 | 14\.88 |
> > >
> > > &nbsp;
> > >
> > > ##### Table 5:
> > > > ResNet-18 CIFAR-10 Test Accuracy under untargeted attack, varying the number of PGD steps. Crafting model trained to 20 epochs instead of 40.
> > >
> > > | PGD Steps: | 20     | 30     | 40     | 50     |
> > > | :--------- | :----- | :----- | :----- | :----- |
> > > | Accuracy (%):  | 33\.42 | 28\.58 | 36\.67 | 24\.94 |
> > >
> > > &nbsp;
> > >
> > > ##### Table 6:
> > > > ResNet-18 CIFAR-10 Test Accuracy under untargeted attack, varying the number of PGD steps. Crafting model trained to 60 epochs instead of 40.
> > >
> > > | PGD Steps: | 20     | 30     | 40     | 50     |
> > > | :--------- | :----- | :----- | :----- | :----- |
> > > | Accuracy (%):  | 53\.03 | 36\.29 | 25\.4 | 26\.37 |
> > >
> > > Additionally, while our method does use more PGD steps in computation, our method does not rely on model retraining during crafting. We have found this is not the case for the loss minimization objective in [2]. To this end, we experiment with the loss minimization objective found in [2] (by simply reversing the sign of the loss found in our untargeted attack). In our framework with a fixed crafting model (with random initialization, 250 PGD steps, differentiable data augmentation, restarts, etc.) poisons generated this way degrade a victim's CIFAR-10 accuracy to only 68.03%. As a side note, this leads to similar computation times even though we use more PGD steps (1:12 vs. 1:16 on 1 GeForce RTX 2080 Ti).
> > >
> > > Does this answer your concerns/do you have any other questions you would like us to address?

---

> > > > ### Comment · Reviewer_7LBn · 2021-08-25
> > > > **Thanks for your response**
> > > >
> > > > Thank you very much for your efforts. The new experiments make the factors somewhat clearer. It turns out that two strong factors are differentiable data augmentation and the number of PGD steps, while the number of training epochs of the surrogate model may be less important. Could you report the clean test accuracies of the surrogate models? They might be also informative.
> > > >
> > > > Possibly I do not understand correctly. I noticed that the results in Table 1 of the new experiments show that the accuracy is 29.19% for untargeted attack with 50 PGD steps and differentiable data augmentation, while in the third column of Table 10 of Appendix A of the submission, the accuracy is 16.53% for the same settings (untargeted attack with 50 PGD steps and differentiable data augmentation). Are the experimental settings of the two tables consistent? If the settings are the same, why their results are different?
> > > >
> > > > Besides, by comparing Table 3 and Table 4 of the new experiments, it turns out that targeted attacks are much more effective than untargeted attacks. Am I right about this? However, in the third column of Table 1 of the submission, untargeted attacks are slightly more effective than targeted attacks (6.25% vs 7.25%). Why is that? Does this contradiction come from the use of differentiable data augmentation?

---

> > > > > ### Author Response · Authors · 2021-08-25
> > > > > **Clarifications**
> > > > >
> > > > > > Could you report the clean test accuracies of the surrogate models?
> > > > >
> > > > > Yes, of course. The average clean accuracy for the surrogate model trained for 20 epochs was 88.67%. For the 40 epochs runs, the average accuracy was 92.51%, and for the 60 epochs model, average accuracy was 93.56%.
> > > > >
> > > > > &nbsp;
> > > > >
> > > > > > Are the experimental settings of the two tables consistent? If the settings are the same, why their results are different?
> > > > >
> > > > > These two settings are slightly different. There are a few factors that lead to the discrepancy in results:
> > > > >
> > > > > * The testing settings are different.  The testing setup we include in our submitted code (the one we use to generate the numbers in the main submission) is a modification of the 3rd party repo that we link to in our original response. We used this stock 3rd party repo for our tables in the response to ensure the results are as “objective” as possible (a question was raised over our testing setup in a previous comment), although we think the modifications we made are quite standard. Note though that the important part is that we compare poisoning methods (main body Table 2) across a fixed testing environment, which we do. For clarification, our testing environment modifies the 3rd party repo in the following ways:
> > > > >
> > > > >     * Training for 100 epochs in our setup vs. 200 epochs.
> > > > >     * Using multi-step scheduler instead of a cosine scheduler for the learning rate.
> > > > >
> > > > >
> > > > > * We generated the poisons in the response tables with only 1 restart. We did this because of computational constraints and our desire to share numbers in a timely fashion. We usually perform 8 restarts during poison crafting.
> > > > >
> > > > >
> > > > > * The variance from run to run of the untargeted attack is greater than our targeted attacks. See the next paragraph for more discussion on this. We note this in Appendix A9 and the README of our submitted code, and we have since updated section 3.3 and 3.4 to include more information on this. Generally, we select the most potent poisons to report results. Because the practitioner in our threat model has access to all data they wish to poison, we believe this is a realistic strategy/assumption. However, for time considerations, we did not do this in the ablations detailed in the response tables.
> > > > >
> > > > > &nbsp;
> > > > >
> > > > > > it turns out that targeted attacks are much more effective than untargeted attacks
> > > > >
> > > > > We see that the untargeted variant *can* be as potent as the targeted variant at “full” capacity (i.e. $\varepsilon = 8/255$, with differentiable data augmentation). This is reflected in the numbers you see in Table 1 (6.25% vs. 7.25%). Note though that these numbers reflect the *best* attacks we found. The variance in attack success from run to run is much higher with the untargeted objective than the targeted objective. We note this in Appendix A9, and hypothesize that a good untargeted attack arises with a well distributed attack matrix. For reference, we do include the best untargeted attack data in our supplementary material, and mention that an attacker may need to run a number of untargeted attacks to find an optimal set of perturbations. In our use case (companies protecting released data), the practitioner would be able to select the optimal poisons. We have amended sections 3.3 and 3.4 (where we initially bring up the relative superiority of our class targeted attack) to reflect the higher variance of untargeted runs.
> > > > >
> > > > > You are correct though that the targeted objective is in general much more potent. We see this in Table 1 with the lower $\varepsilon$ bounds. For example, at $\varepsilon = 4/255$, the untargeted attack degrades a ResNet-18’s accuracy to 40.14%, whereas the targeted attack degrades it to 23.79%. Similar behavior is seen on ImageNet where the targeted attack degrades victim accuracy to 1.45% whereas the untargeted attack results in a victim accuracy of 36.63% (Table 3). We discuss this discrepancy in section 3.4, and we believe this arises from the more complicated label space of ImageNet, where achieving a well distributed attack matrix is more difficult. In terms of the role of differentiable data augmentation (DDA), we believe this is very important for the success of the untargeted attack, but not important for the success of the targeted attack. Apropos the last part of the rolling discussion, we have since updated the main body to reflect the findings on DDA’s and PGD step importance to untargeted attacks. We again thank you for inspiring these findings.
> > > > >
> > > > >
> > > > > Does this clarify your concerns?

---

> > > > > > ### Comment · Reviewer_7LBn · 2021-08-26
> > > > > > **Thanks**
> > > > > >
> > > > > > Thanks for the detailed clarifications.
> > > > > > I am still a bit confused about the discrepancy since it is not negligible.
> > > > > >
> > > > > > > We generated the poisons in the response tables with only 1 restart. We did this because of computational constraints and our desire to share numbers in a timely fashion. We usually perform 8 restarts during poison crafting.
> > > > > >
> > > > > > Does this mean that the number of restarts is another important factor that has been overlooked? In particular, performing 8 restarts instead of 1 restart would degrade the accuracy from 29.19% to 16.53%. Is that the case?

---

> > > > > > > ### Author Response · Authors · 2021-08-26
> > > > > > > **Clarification on number of restarts**
> > > > > > >
> > > > > > > Thank you for your continued dedication to the review process - this is much appreciated in the tumultuous environment of NeurIPS!
> > > > > > >
> > > > > > > &nbsp;
> > > > > > >
> > > > > > > > Does this mean that the number of restarts is another important factor that has been overlooked?
> > > > > > >
> > > > > > > Thank you for the question. Number of restarts is a difference between the two settings where the numbers were generated, and thus we listed it as a possible source of the difference. While it may contribute, we do not think it is a significant factor in the discrepancy. Since you brought it up though, we are now running ablations on the number of restarts and will include them in the updated appendix, but this is a more time intensive process as N restarts essentially costs Nx the amount of time. However, we have run a small number of trials to further explain this discrepancy, and have included them below. Here are the takeaways:
> > > > > > > * We find that the main cause of discrepancy is most likely the variance of untargeted attacks. Again, we believe this is not an issue for our threat model as the attacker could simply perform several runs, and select the one that degrades accuracy the most. We did not do this for the runs in the previous response tables as we wanted to get results to you as quickly as possible. It looks like we were somewhat unlucky with the run we performed for these experiments! If we had selected the best run out of the 4 we have since generated (18.09% in Table 7), the discrepancy would be much less significant. As stated previously, although we include discussion of the variability of the untargeted attack in Appendix A9, from your comments, we agree that this is a point of confusion that should be clarified in the main body. Because of this, we have changed our manuscript to better reflect this, and will include error bars not just for the validation runs on a fixed poison set, but also poison generation.
> > > > > > >
> > > > > > > * More restarts slightly increase the average potency, and seem to decrease the variability compared to the single restart runs, but the difference is quite small, and it's hard to tell for a small number of runs. However, we will include a table in the appendix ablating the number of restarts to assuage concerns over this being a crucial factor. These runs will take longer to finish, though.
> > > > > > >
> > > > > > > &nbsp;
> > > > > > >
> > > > > > > ### Experimental details
> > > > > > > We use the setting of the previous response Table 1. However, because restarting crafting requires more computation, we only focus on the 50 PGD step column, as this is the number about which you raised concerns. Then, we craft poisons in two frameworks. Framework 1 (Table 7) consists of 4 runs of poison generation, varying the model/poison initialization. Framework 2 (Table 8) consists of  4 runs of poison generation, varying the model/poison initialization, but with 4 restarts each. Here are the results for these poisons:
> > > > > > >
> > > > > > > &nbsp;
> > > > > > >
> > > > > > > #### Table 7:
> > > > > > > > ResNet-18 CIFAR-10 Test Accuracy under untargeted attack, with 50 PGD steps, DDA, and 1 restart.
> > > > > > >
> > > > > > > |Run number: | run 1     | run 2     | run 3     | run 4     |
> > > > > > > | :--------- | :----- | :----- | :----- | :----- |
> > > > > > > | Accuracy (%):  | 19\.36 | 32\.80 | 23\.88 | 18\.09 |
> > > > > > >
> > > > > > > **Average**: 23.53%
> > > > > > >
> > > > > > > &nbsp;
> > > > > > >
> > > > > > > #### Table 8:
> > > > > > > > ResNet-18 CIFAR-10 Test Accuracy under untargeted attack, with 50 PGD steps, DDA, and 4 restarts.
> > > > > > >
> > > > > > > |Run number: | run 1     | run 2     | run 3     | run 4     |
> > > > > > > | :--------- | :----- | :----- | :----- | :----- |
> > > > > > > | Accuracy (%):  | 25\.50 | 19\.85 | 23\.50 | 21\.67 |
> > > > > > >
> > > > > > > **Average**: 22.63%
> > > > > > >
> > > > > > > &nbsp;
> > > > > > >
> > > > > > > Does this answer your concern?

---

> > > > > > > > ### Comment · Reviewer_7LBn · 2021-09-02
> > > > > > > > **Thanks for the response**
> > > > > > > >
> > > > > > > > Thanks for further experiments and sorry for my late reply. I am glad to see that my suggestions helped to identify the contributing factors for attack effectiveness.
> > > > > > > > It turns out that at least three strong factors emerge during the rebuttal:
> > > > > > > >
> > > > > > > > 1. the variance of the attack effectiveness itself
> > > > > > > > 2. the use of differentiable data augmentation
> > > > > > > > 3. the number of PGD steps
> > > > > > > >
> > > > > > > > I would like to suggest the authors to reflect these key factors (and other minor factors) more clearly in the main body, since they are important points that can differentiate this work from previous work. Surely, achieving SOTA performance itself is a good contribution, but for this venue, it alone may not be enough. In my opinion, another contribution of this work would be to sufficiently clarify why prior implementations of adversarial examples cannot achieve SOTA.
> > > > > > > >
> > > > > > > > There are some remaining mysteries summarized as follows.
> > > > > > > >
> > > > > > > > 1. I still feel a bit confused about the variance of untargeted attacks. If the variance is as large as Table 7 in the rebuttal, why the variance reported in Table 1 of the submission is so small? I would like to suggest the authors to report the results such as Table 7 for both targeted and untargeted attacks in the revision to clarify the variance from run to run of the attacks.
> > > > > > > > 2. Another confusion comes from the use of differentiable data augmentation. The results of Table 1 and Table 3 in the rebuttal clearly show that differentiable data augmentation is a strong factor. The authors also acknowledged this. However, I found that it was shown in the submission that the factor is minor by comparing the third row ("SIGNADAM W/ AUG") and the fourth row ("SIGNADAM W/O AUG") in Table 11 of the appendix. The authors may want to make a clarification.
> > > > > > > > 3. Finally, I still thought that it would be better if the authors could plot the data in Table 10 using curves, to show the trend of the decreasing accuracy as the number of PGD steps increases (e.g., plotting every 50 steps, up to convergence, say, 1000 steps). This would show the limits of the poisons. Otherwise, the choice of crafting poisons with 250 steps of PGD is not well-motivated. Readers may feel that 1000 or more steps would be better.

---

> > > > > > > > > ### Author Response · Authors · 2021-09-02
> > > > > > > > > **Thank you for the response**
> > > > > > > > >
> > > > > > > > > Thank you for your response.
> > > > > > > > >
> > > > > > > > > >It turns out that at least three strong factors emerge during the rebuttal:
> > > > > > > > >
> > > > > > > > > Yes, it does indeed appear that those are the factors that are key for a strong untargeted attack. We agree that this is important to make clear in the main body, especially because it shines new light on a technique that was previously thought to fail (as in [2]) and so we have added a subsection after 3.2 in our original manuscript entitled “Technical Components for Strong Attacks'' where we discuss these three primary issues, as well as mentioning restarts and crafting model epochs. In this section, we include references to updated/new tables where these issues are thoroughly investigated, and make clear which details affect untargeted attacks more than targeted attacks. We thank you for these suggestions and we sincerely believe your feedback has improved the presentation of our work!
> > > > > > > > >
> > > > > > > > > &nbsp;
> > > > > > > > >
> > > > > > > > > ### On the remaining mysteries
> > > > > > > > >
> > > > > > > > >
> > > > > > > > > &nbsp;
> > > > > > > > >
> > > > > > > > > > If the variance is as large as Table 7 in the rebuttal, why the variance reported in Table 1 of the submission is so small?
> > > > > > > > >
> > > > > > > > > The variance in Table 1 is taken over the *victim training* runs. So when a fixed set of poisons is generated, we trained a victim several times and reported the variance of these runs. The variance we see in Table 7 is over *crafting* runs where the attacker crafts multiple sets of poisons. Thank you for bringing this confusion to our attention. We have made this clear in the updated main body tables.
> > > > > > > > >
> > > > > > > > >
> > > > > > > > >
> > > > > > > > > &nbsp;
> > > > > > > > >
> > > > > > > > > > … (Results in)  in Table 11 of the appendix
> > > > > > > > >
> > > > > > > > > Thank you for bringing this confusion up. It has been some time since we ran the experiments for these Appendix Tables, but we believe that the two SIGNADAM numbers were generated with our targeted attack. This is an oversight on our part as the descriptions in the table do not make it clear from what poison set these numbers are generated. Because we introduce two variants of the attack, we need to be very clear about the setting of each number. We apologize for this confusion and we have updated this Table to include clearly labeled results for both targeted and untargeted attacks.
> > > > > > > > >
> > > > > > > > >
> > > > > > > > > &nbsp;
> > > > > > > > >
> > > > > > > > > > Finally, I still thought that it would be better if the authors could plot the data in Table 10 using curves ...
> > > > > > > > >
> > > > > > > > > We think this is a good idea and will present the data in this form. Thank you again for this suggestion. We put results in tables for the response period simply because of the markdown formatting. We agree though that going further out in PGD steps is also important and we are currently filling in this table. We do suspect that, as with evasion attacks, there are diminishing returns going further with PGD steps. The attacks almost always end up in a “corner” of the $\ell_\infty$ ball, which leads us to believe that the optimization stabilizes there, and would not result in a different attack had we done more PGD steps.
> > > > > > > > >
> > > > > > > > > &nbsp;
> > > > > > > > >
> > > > > > > > > Does this answer your concerns?

---

### Official Review · Reviewer_gJ9i · 2021-07-16

**Rating:** 7
**Confidence:** 3

**Summary:**

This paper investigates adversarial examples as a data poisoning method and offers insights as to why they are effective. The idea of using adversarial examples for poisoning is motivated by the intractability of standard data poisoning (cast as a bi-level optimization problem) for neural networks. The paper focuses on _availability attacks_ where the goal of poisoning is to maximize the expected loss of the model. A suite of experiments demonstrates the effectiveness of adversarial examples for poisoning. Experiments include: comparisons with baselines on CIFAR-10 and a facial recognition application; testing attack transfer to different model architectures; testing effectiveness on ImageNet (as a large-scale setting); and testing effectiveness for different poisoning ratios. Seven defenses are also tested, and the only effective defense is found to be adversarial training. Experiments are also run to test the authors’ hypothesis that adversarial examples are effective because they contain discriminative features for the “wrong” class.

**Limitations And Societal Impact:**

Yes, this is covered adequately in Sections 3.1 and 5 of the paper.

**Main Review:**

This paper is based on a very simple idea: seeing whether a well-known test-time attack (adversarial examples) is effective as a training-time attack. There is essentially no work required to adapt the attack to a training-time setting, however the authors do propose a novel _class targeted_  variant for which the class labels are permuted. Despite the simplicity of the idea – I see it as an obvious attack to try – it does not appear to have been investigated in the literature. The only prior work I could find is the following (not cited):

> CHAN-HON-TONG A. An Algorithm for Generating Invisible Data Poisoning Using Adversarial Noise That Breaks Image Classification Deep Learning. Machine Learning and Knowledge Extraction. 2019; 1(1):192-204. https://doi.org/10.3390/make1010011

Although the paper lacks new technical ideas, I think the empirical study is well done and offers important insights that ought to be shared with the community. I was surprised by the effectiveness of the approach, especially given its relative simplicity compared to existing approaches for poisoning neural networks. There are very few stones left unturned in the experiments – the attack is tested on multiple datasets (including at larger scales), transferability is assessed for different network architectures, defenses are tested, and there are further experiments testing the effect of various parameters in the appendices. It would have been interesting to see how the compute time varies for each type of attack, however this would be difficult to measure fairly on heterogeneous hardware.

For an empirical paper such as this one, I think it’s important that clearly documented scripts and code be made available to support reproducibility. I have not been able to carefully examine the repository included in the supplementary material, however the README in the root directory seems light on detail.


**Time Spent Reviewing:**

3

---

> ### Author Response · Authors · 2021-08-10
> **Response to Reviewer gJ9i**
>
> Thank you for your time spent reviewing and for your constructive comments.
>
> &nbsp;
>
> ### Responses to specific concerns:
>
>
> &nbsp;
>
> > The only prior work I could find is the following (not cited):
>
> Thank you for pointing out this work to us. We have since added a reference to this work in Section 2 of our manuscript. This work does propose using loss maximization to degrade accuracy of a victim model, but it does not seem like they perform a class targeted attack. We hypothesize the relative superiority of our results might stem from using PGD (as opposed to FGSM) and differentiable data augmentations. Our work also uniquely demonstrates this idea as a practical attack on deep learning systems with extensive ablations, and tests on modern datasets and architectures.
>
> &nbsp;
>
> > It would have been interesting to see how the compute time varies for each type of attack
>
> Thank you for your compliment on the thoroughness of our experiments. As for the time required for each attack, this is highly dependent on the number of restarts the poisoner deploys, and the number of PGD steps. On CIFAR-10, crafting can take less than one hour with a single NVIDIA GeForce RTX 2080 Ti Graphics Card. This constitutes roughly the same amount of time as crafting an evasion attack, but for the entire training dataset.
>
>
> &nbsp;
>
>
> > For an empirical paper such as this one, I think it’s important that clearly documented scripts and code be made available to support reproducibility
>
> We have now made the README more detailed. Thank you for the suggestion. If you have specific questions/things you want to run, we are more than happy to clarify.

---

### Official Review · Reviewer_RuBF · 2021-07-18

**Rating:** 4
**Confidence:** 5

**Summary:**

This paper proposes a method for poisoning the train dataset of a victim classification model. Even if the poisoner does not know any information about the target victim model training procedure, it can still seriously degrade the classification accuracy of the victim model on clean data. This is done by replacing total or partial training data as corresponding adversarial examples (AEs) with ground truth labels. It is verified that AEs with ground truth labels indeed degrade the classification performance of models.

**Main Review:**

Originality: This paper contributes some new ideas.
Clarity: The organization of this paper needs to be improved. Some experimental details are presented in the technical part, making it difficult to capture the core idea of the technical contributions.
Significance: The problem formulation of this paper is almost the same as that of the work [1]. Although this paper utilizes a surrogate model to approximately solve the problem, the work [1] attacks DNNs by back-gradient without any surrogate model, which is more challenging. Therefore, the novelty or improvement of the proposed work is rather limited.
Weaknesses and my questions about this paper:
1.	The experiment of this paper is not sufficient. Firstly, there is no comparison with other data poison methods, especially with [1], which is very similar to the proposed one.
2.	This work utilizes existing attack methods on a surrogate model. It is similar to use the transferability of adversarial examples directly. The author needs to further claim the novelty and contribution of the proposed method.
3.	The proposed method might be invalid when adversarial detections are involved. More precisely, the defender can utilize existing detection methods, such as like LID[2], MD[3], and KB[4], to remove those poisoned examples. Thus, there should be some tests on evaluating the robustness of the proposed method against adversarial detections.
4.	The author has pointed out that their method performs unsatisfactorily to the defense of adversarial training techniques. In fact, such an limitation is fatal as  the adversarial training is not so expensive as the authors claimed. Some adversarial training method like FastAdv [5] improves the training speed significantly.





**Time Spent Reviewing:**

2 hours

---

> ### Author Response · Authors · 2021-08-03
> **Please include references**
>
> Thank you for your review. We will respond more thoroughly soon, but we noticed it seems like you're citing works whose references aren't included in your review (e.g. [1]). Please include these when you get a chance so we can respond!

---

> ### Author Response · Authors · 2021-08-10
> **Response to Reviewer RuBF**
>
> Thank you for your time and thoughtful comments. Below we respond to concerns you raise.
>
> &nbsp;
>
> ### On the weaknesses you describe:
>
> &nbsp;
>
> The reviewer claims there are similarities between our paper and a paper labeled as [1].  We previously posted asking for the reference [1] to be provided, but we are currently unaware of what paper that is referring to.  We are eager to respond to your claim if we are provided details of that work you are referring to.
>
> > there is no comparison with other data poison methods ...
>
> We agree that comparing to other poisoning methods is important, which is why our original submission compared to 4 state of the art availability attacks (e.g., poisoning on the whole dataset) on deep networks, including:
> * [TensorClog](https://ieeexplore.ieee.org/document/8668758)
> * [Gradient Alignment](https://arxiv.org/abs/2103.02683)
> * [Unlearnable Examples](https://arxiv.org/abs/2101.04898)
> * [DeepConfuse](http://129.211.169.156/publication/neurips19deepc.pdf)
>
> Please see Tables 2, 4.
>
> While there are other types of data poisoning attacks, like backdoor attacks, targeted poisoning, and subpopulation attacks, these have different threat models and objectives than our domain of interest, which is availability poisoning for the purpose of secure dataset release.
>
> &nbsp;
>
> > It is similar to use the transferability of adversarial examples directly
>
> While the transferability of adversarial examples as *evasion* attacks is a well studied topic, it does not tell us one way or another whether adversarial examples crafted on a surrogate model will transfer as *poisons* to a new victim model. In order to study this effect, we conduct transferability experiments both across random initializations, and across a variety of commonly used architectures. See Table 1 for results transferring across new (black-box to the poisoner) initializations, and Appendix Table 12 for transferability across network architecture. Modifications to our algorithm to improve black box transferability would certainly be an interesting direction for future research.
>
> &nbsp;
>
> > The proposed method might be invalid when adversarial detections are involved
>
> The reviewer did not provide information about what papers are being referred to, but we will try our best to respond to this criticism in the absence of a complete review.
> The main application area for our algorithm is one wherein a company poisons data before releasing it to the public so that an actor who scrapes said data cannot train a useful model on that data. In this scenario, if a defender detects all the adversarial examples and removes them, then they will be left without any data scraped from the company’s platform since all released data is perturbed, and the company will have achieved its goal. The goal of a company deploying our method is to prevent competitors from using scraped data to train a model. If the competitor "detects" and throws out poisoned data, that is certainly a positive result for the company deploying our method.
>
> &nbsp;
>
> > The author has pointed out that their method performs unsatisfactorily to the defense of adversarial training techniques ...
>
> Our method does indeed perform suboptimally when the scraping actor uses adversarial training. But we stress that this is not a unique disadvantage to our method as previous state of the art availability attacks also suffer from this (see Unlearnable Examples and DeepConfuse). You are correct though that there exist faster adversarial training techniques like FastAdv, or Adversarial Training for Free, so we have updated Section 3.7 to reflect this and added these citations. Furthermore, we have added entries in Table 7 for this approach. Thank you for pointing these out.
>
> However, as we state in Section 3.7, forcing the scraping actor to adversarially train is still quite beneficial to the company deploying our algorithm as adversarial training significantly degrades natural accuracy (for example, a drop of nearly 30% on ImageNet), which was the company's goal to begin with.

---

### Decision · Program_Chairs · 2021-09-27

**Decision:**

Accept (Poster)

**Comment:**

This paper presents a technique	to poison datasets with	adversarial examples.
This an	interesting and	new direction, and the results are (often order of magnitude)
more effective than prior work that relied on much more	sophisticated techniques.

While the reviewers are	concerned about	the tone of the	paper over-claiming in
its originality (and I would encourage the authors to compare to the prior work
the reviewers have pointed out), the results are sufficiently strong to	merit
publication.